# Multidecadal, continent-level analysis indicates agricultural practices impact wheat aphid loads more than climate change

Xiao Sun [1], Yumei Sun[1], Ling Ma[1], Zhen Liu[1], Qiyun Wang[1], Dingli Wang[1], Chujun Zhang[1], Hongwei Yu[1], Ming Xu[2,3], Jianqing Ding [1✉] & Evan Siemann[4]

Temperature has a large influence on insect abundances, thus under climate change, identifying major drivers affecting pest insect populations is critical to world food security and agricultural ecosystem health. Here, we conducted a meta-analysis with data obtained from 120 studies across China and Europe from 1970 to 2017 to reveal how climate and agricultural practices affect populations of wheat aphids. Here we showed that aphid loads on wheat had distinct patterns between these two regions, with a significant increase in China but a decrease in Europe over this time period. Although temperature increased over this period in both regions, we found no evidence showing climate warming affected aphid loads. Rather, differences in pesticide use, fertilization, land use, and natural enemies between China and Europe may be key factors accounting for differences in aphid pest populations. These long-term data suggest that agricultural practices impact wheat aphid loads more than climate warming.

[1] State Key Laboratory of Crop Stress Adaptation and Improvement, School of Life Sciences, Henan University, Kaifeng 475004, China. [2] Key Laboratory of Geospatial Technology for the Middle and Lower Yellow River Regions (Henan University), Ministry of Education, Kaifeng 475004, China. [3] The College of Geography and Environmental Science, Henan University, Kaifeng 475004, China. [4] Department of Biosciences, Rice University, Houston, TX 77005, USA. ✉email: jding@henu.edu.cn

Food security and agricultural ecosystem health are of crucial concern throughout the world[1,2]. Pest insects, however, have been causing large losses to crop yields since crop domestication about ten thousand years ago[3,4]. Applications of pesticides and fertilizers are important for improving yields, however, over-use of synthetic insecticides and fertilizers may have negative impacts on agricultural sustainability and human health[5,6]. Moreover, pest populations may outbreak more frequently and more intensely under climate change[7–9] posing a greater threat to global food security. Wheat (*Triticum aestivum* L.) is one of the most important cereal crops[10], providing 20% of the calories consumed by humans in the world[11]. Thus, developing healthy agroecosystem management in wheat could substantially improve global food security and food quality.

Climate change has been identified as an important factor affecting pest insect abundances directly[12–14] and indirectly via altering host plant-pest-natural enemy interactions[15,16]. While many studies have shown that elevated temperature has increased pest insect abundances[17,18], some others reported that climate warming may decrease insect individual development and population growth rates[19]. These inconsistent findings suggest that the effects of climate warming on insects are complex[20], likelyF due to other factors (e.g., plant nutrients, plant defenses, and natural enemy dynamics) that are associated with insect development and population dynamics that could interact with each other and with the warming climate[21,22]. Therefore, the numerous experiments conducted to date with a single factor design (e.g., simulated warming) or short-term field monitoring (e.g., months or several years) at the local or regional scale are unlikely to unveil the mechanisms driving long-term or large-scale pest loads. Rather, studies at broad geographic scales with long-term field data are needed to tease apart the role of climate warming from other factors in regulating pest insect populations under natural conditions[23,24].

Synthetic chemical insecticides have been used worldwide for pest management for more than a century[25,26], which has greatly increased agriculture yields. Nevertheless, extensive usage of pesticides can lead to high levels of resistance to insecticides[27,28], which further complicates aphid control[29]. Alternatively, improper use of pesticides has also negatively affected biodiversity in agroecosystems[30], and can lead to pest outbreaks by reducing biocontrol potential[31,32], as natural enemies are a key biotic factor that suppresses pest insect densities[33]. In fact, these natural enemies are also affected by land use because diverse landscape composition is beneficial to biodiversity which could enhance ecosystem services that suppress pest abundances by increasing natural enemies[34]. Alternatively, landscape simplification or agriculture intensification could threaten biodiversity and natural habitats[35], resulting in declining ecosystem services for pest biocontrol, thus increasing pest abundance and insecticide and fertilizer application[36]. In this context, combining and synthesizing long-term data of pest insects, climate change, insecticide, and fertilizer inputs, as well as land use, could provide insights into agroecosystem management and decision making[37].

Overuse of chemical fertilizers not only threatens the health of agroecosystems, but could induce pest insect outbreaks as well[38]. Nitrogen fertilizer applications have been considered to be positively associated with some insect populations by improving plant nutrition that enhances herbivorous insect development[39]. For example, aphids, one of the most devastating insects in agriculture, are particularly sensitive to nitrogen content in crop plants, thus applications of nitrogen fertilizers have often led to wheat aphid outbreaks[40]. Thus, in addition to the effects on crop yields and agroecosystems, consideration of the roles of chemical fertilizer applications in regulating pest insect populations is also critical to developing effective agroecosystem management systems.

Agricultural intensification since 1950 has resulted in serious loss of biodiversity and ecosystem function within agricultural landscapes[41]. Negative effects of the proportion of cultivated land were found on biological control by natural enemies[42]. A relative increase of cultivated land from 2 to 100% in the 1 km radius reduced the level of natural pest control by about 46%, suggesting that landscape is a major determinant of pest control in agroecosystems[43,44]. In contrast, low-intensity agriculture enhanced biodiversity and promoted biological control[45]. To satisfy the food demand of vast populations in China, agricultural fields have long been intensified and landscapes simplified[46], likely decreasing ecosystem services in wheat.

China is one of the most important wheat producers in the world, planting over 10% of the global area in 2016, while Europe is a major wheat producer in the world with ~25% of the world sowing area (https://www.fao.org/faostat)[11]. The wheat production areas in China and Europe share similar latitudes and temperate climates, however, differences in insecticide and N fertilizer applications, as well as different land use practices between these two regions may result in different wheat aphid loads. Therefore, comparing the wheat aphid populations and agricultural practices (pesticides, N fertilizer, and land use) between China and Europe may provide new insight into the understanding of pest populations at the continental scales under climate change. In this study, we aim to determine the main drivers of aphid dynamics by meta-analyses across the two continents (12 provinces which represent over 95% of the national wheat production in 2017 in China and 10 countries which account for over 50% of Europe wheat production in 2017 in Europe) over a 47-year period from 1970 to 2017.

With this rich data set, here, we explore how abiotic or biotic factors jointly and quantitatively drive wheat aphid population dynamics in agroecological systems at large spatio-temporal scales. Specifically, we asked the following questions: (i) how have wheat aphid loads varied in China and Europe over the past several decades? (ii) do these temporal patterns differ between China and Europe? (iii) are these patterns associated with climate warming, land use, fertilization, and/or application of pesticides? We hypothesized that: (i) relative to Europe, increasingly high use of pesticides and N fertilizers together with agriculture-dominated landscapes (and reduced control by natural enemies) in China will drive increased wheat aphid loads, and (ii) as climate in the wheat growing regions is similar between China and Europe, changes in temperatures (warming) will have comparable impacts in the two regions. Our study with long-term data obtained from multiple locations in China and Europe showed agricultural practices impact wheat aphid loads more than climate warming. Our findings could highlight the drivers of aphid population dynamics in long term series, can help promote agroecological health by maximizing ecosystem service benefits and minimizing agrochemical inputs under global change.

## Results

**Abundance of wheat aphids**. Wheat aphid loads increased significantly from 1970 to 2017 in several ten-day periods (mid-March, late-March, early-April, mid-April, early-May) in China (Fig. 1a–i). However, the loads of wheat aphids did not vary over this time period from May to July in Europe (Fig. 1j–r). Over the last five decades, aphid loads increased overall and in each part of the growing season in China with the most dramatic increases earlier in the growing season. In contrast, aphid loads decreased in Europe with this pattern overall and in the early and middle of the growing season significant (Fig. 2). The patterns for China and Europe differed overall and in the early and middle of the growing season (i.e., the 95% CI did not overlap). The funnel plot

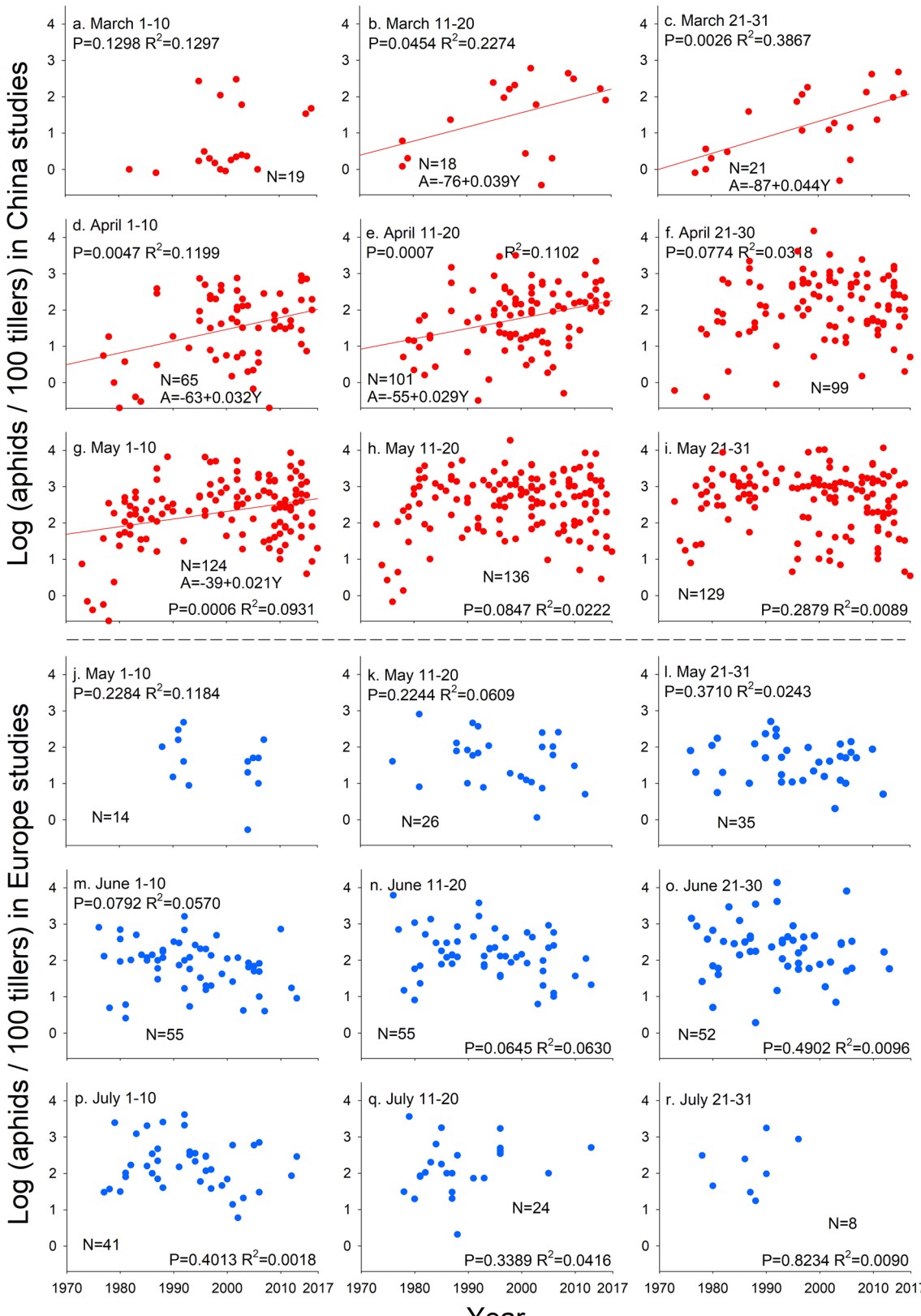

**Fig. 1 Historical wheat aphid loads in China and Europe by season.** Wheat aphid (log transformed) loads in different times of the year in China (**a–i**) and Europe (**j–r**) during 1970–2017. P and $R^2$ values are from regressions. Lines indicate significant linear relationships between aphid loads and time. Including a random term for paper or for province (China) or country (Europe) makes the relationship in **b** not significant but the other significant relationships are unaffected.

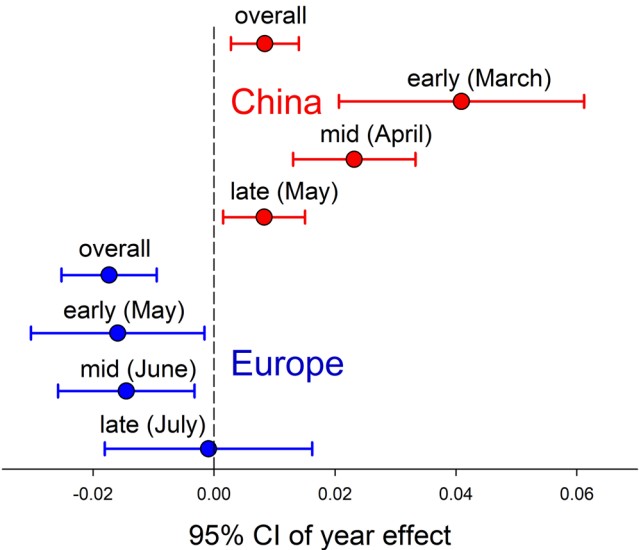

**Fig. 2** The temporal change in early, mid and late season wheat aphid loads (log transformed) from 1970 to 2017 in China and Europe (95% confidence interval of slopes).

and Egger's test ($P = 0.3892$) indicated that publication bias did not affect these results (supplementary Fig. 1).

**Abundance trend of natural enemies coupled with wheat aphids**. Natural enemy loads of 499 data points covering 11 provinces decreased significantly in some ten-day periods (late-April, mid-May) over years from April to May during 1980–2017 in China (Supplementary Fig. 2). Wheat aphid loads and natural enemy abundances showed opposite patterns over time in China with aphids increasing (slope = +0.0117, $P < 0.001$) and enemies decreasing (slope = −0.176, $P < 0.001$). Fewer data were available for Europe. We collected natural enemy loads of 108 data points covered 5 countries during 1992–2007 in Europe. The results showed that aphids decreased (slope = −0.0099, $P = 0.037$) and natural enemies increased (slope = +0.458, $P < 0.001$) in Europe (data shown in Fig. 3a, b). Specifically, there was a trend for ladybirds and hoverflies to decrease in China from 1980 to 2017 (Supplementary Fig. 3a) and increase in Europe from 1990 to 2008 (Supplementary Fig. 3b).

**Changes in inputs of pesticides and N fertilizer over time**. N-fertilizer used per area of wheat in China increased almost two and a half fold from 1980 to 2000 (Fig. 3c) and pesticide inputs rose sharply in the period of 1990 to 2015 (Fig. 3d) in China, while aphid loads increased (Fig. 3a) and natural enemy abundances decreased (Fig. 3b) during this same time period. In recent years, there has been a trend for N-fertilizer use to reduce (Fig. 3c) and pesticide use to stabilize (Fig. 3d) in China.

In Europe, N fertilization per area cultivated has been stable, as seen in N-fertilizer applications in Europe cropland, UK cereals or EU cropland (Fig. 3c). In Europe, total insecticide inputs sharply decreased in the period of 1990–2000, then they were stable from 2000 to 2015 (Fig. 3d), showing a positive relationship between aphid abundance and insecticide use (i.e., both decreased).

**Changes in temperature over time**. The temporal trends in annual average temperature were similar between China and Europe (Fig. 4 and Supplementary Fig. 4). Mean monthly temperature exhibited an increasing trend mainly from 0.01 to 0.08 °C over years among early, mid and late parts of the wheat growing season in

China and Europe during the 46-year period (Supplementary Fig. 5 and Supplementary Data 2). The rates of temperature change (overall) were not correlated to the rates of change in aphid loads for provinces in China ($F_{1,9} = 0.2$, $P = 0.64$) or countries in Europe ($F_{1,5} = 0.1$, $P = 0.80$).

**Land use variation**. The proportion of area under cultivation was high (more than half of the area) in the main wheat producing provinces in China and overall the intensity of agricultural land use was higher in China than in countries in Europe from 1979 to 2015 ($F_{1,19} = 72.1$, $P < 0.001$, Fig. 5). The proportion of area under cultivation increased significantly for seven provinces in China and one country in Europe and it decreased significantly for three provinces in China and six countries in Europe (Fig. 5). A few provinces in China had high proportions of land area under wheat cultivation but overall provinces in China and countries in Europe did not differ significantly ($F_{1,19} = 2.47$, $P = 0.130$). The proportion of land under wheat cultivation increased significantly for two provinces in China and six countries in Europe and it decreased significantly for eight provinces in China and one country in Europe (Supplementary Fig. 6).

## Discussion
Our study with long-term data obtained from multiple locations in China and Europe showed distinct patterns of wheat aphid loads between these two regions, with a significant increase in China but significant decrease in Europe during 1970–2017. During this period, even though mean monthly temperature exhibited an increasing trend over years for most of the wheat growing season in China and Europe, we found no evidence showing climate warming was the key factor affecting aphid population dynamics. Rather, agricultural practices might have contributed to these trends in aphid populations and their variation between regions, among which, pesticides, N fertilization, intensity of land use and natural enemy abundance seemed to be key factors and differences of the effects of any of these factors between China and Europe might account for the variations (Fig. 6).

Temperature has been identified as an important abiotic factor that can directly affect insect abundance[21,47]. An increasing number of studies have found that experimental warming in early summer increases growth rates and/or abundance of wheat aphids over the short-[48] or long-term[49]. Warmer temperature in winter could increase aphid mortality and reduce reproductive potential by reducing nutritional reserves during the dormant diapause period[50]. Across a 20-year period, warmer temperatures in winter affected the emergence of both an aphid and its Hymenopteran parasitoid[51]. In contrast, some experiments with manipulated warming found no evidence that warming increased aphid growth or abundance over the short-[52] or long-term[53], consistent with a study showing no link between mild winters and aphid outbreaks across 26 years[54]. Similarly, in our study, global warming appears not to be an important explanation for the increase of the aphid populations in China when considering the similar climate patterns in Europe where there was no such increase in wheat aphids and that more rapidly warming countries or provinces did not have more rapid increases in aphid loads. Overall, our study, together with previous reports, may suggest that results from short-term field data and warming experiments may not necessarily reflect the effects of climate warming on pest insects over a long-term scale. The responses by individual aphid species to climate warming could be connected through interactions with many other biotic and abiotic factors, subsequently affecting aphid development, reproduction, overwintering, life cycles, and population dynamics[55]. We should note

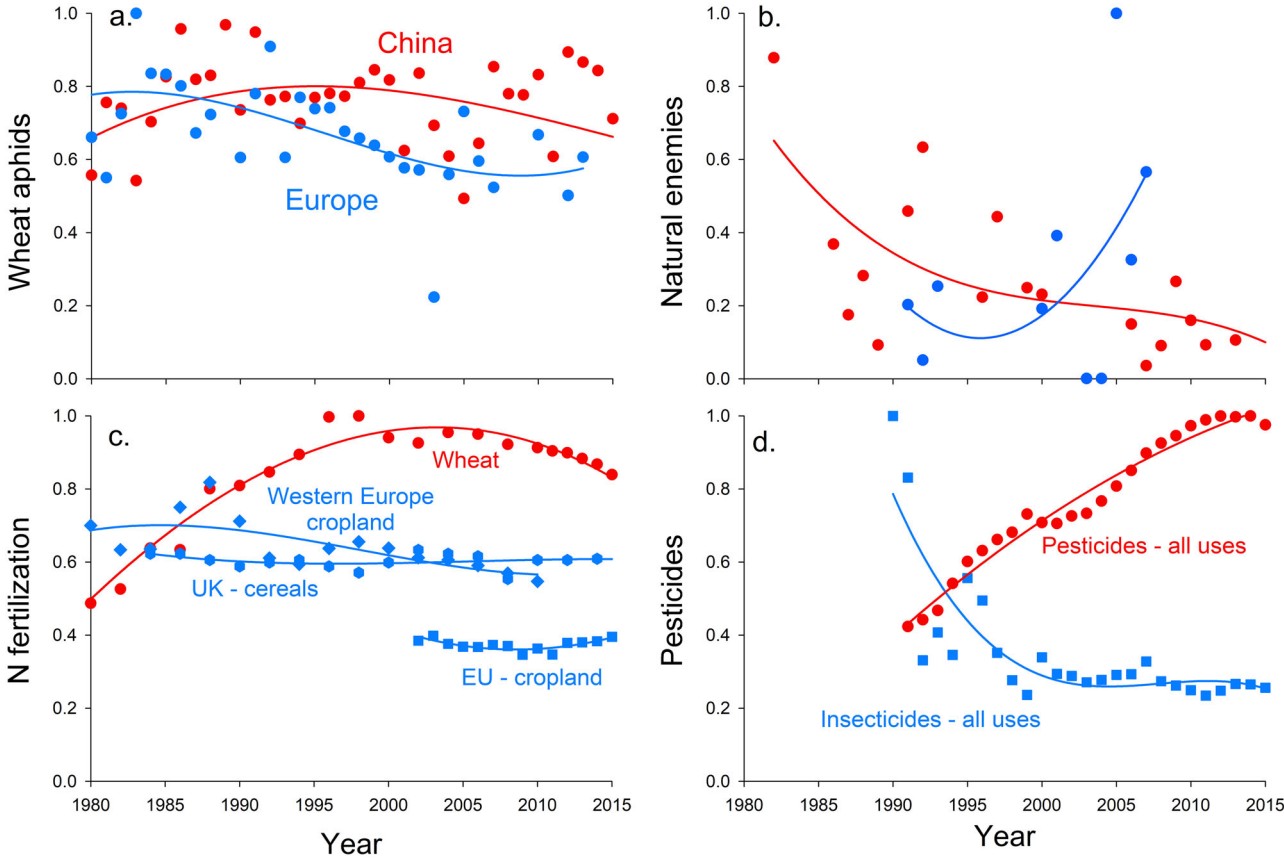

**Fig. 3 Wheat aphids, natural enemies and agricultural practices in China and Europe.** Trends of wheat aphid loads (**a**), natural enemies (**b**), nitrogen fertilizer inputs per area (**c**), and insecticide or pesticide inputs (**d**) in China and Europe from 1980 to 2015. All values are normalized so that the highest value is set at 1. Normalized values (value of 1) are: aphids = 10^4.49 100 tillers$^{-1}$ (i.e., log[aphids] = 4.49 is set to 1); natural enemy loads = 63.5 100 tillers$^{-1}$; N fertilization = 289 kg ha$^{-1}$ yr$^{-1}$; pesticide use [China] = 1.8MT yr$^{-1}$, insecticide use [Europe] = 5kT AI country$^{-1}$ yr$^{-1}$. Curves are third order polynomials fit to illustrate general patterns and not for statistical tests.

that wheat and aphid phenologies differed from March to May in China and from May to July in Europe. Thus, potential effects of aphid phenology on aphid temporal patterns across years should be considered.

The consistent increase of wheat aphid loads in China in the last five decades might be due in part to the consistent decrease of natural enemies in the fields. In fields, there are a range of specialist and generalist natural enemies attacking wheat aphids, including syrphids, lacewings, coccinellids, hemipteran predators, carabids, spiders, and hymenopterous parasitoids[56]. Natural enemy exclusion experiments have shown that aphid populations were depressed markedly by 30–100% on cereal crops when natural enemies were present[57,58]. Conversely, loss of these natural enemies will presumably result in increased aphid populations[59–62]. Therefore, in contrast with the pattern in China we found, the decreasing populations of the aphids in Europe might be related to impact of natural enemies on the decadal scale[63,64]. However, recent reports showed that widespread declines in arthropod biomass and abundance in Europe[65], especially in grasslands and forests[66]. This might because farmlands experienced more intense disturbance than the other land types. Together, the contrasting trends of aphid populations between China and Europe may in part reflect their distinct patterns of natural enemies during 1970–2017.

China is one of the largest consumers of agricultural pesticides in the world today[67]. The consistent decrease of natural enemies in wheat in China might be side-effects of overuse of chemical insecticides that could kill natural enemies[68]. The temporal data

here for China are pesticides overall, not insecticides, and we are assuming that insecticide applications are positively correlated to pesticide applications[67,69]. A series of highly-toxic insecticides have been used extensively in wheat field in China in last decades, including deltamethrin, methomyl, omethoate, fluoroacetamide, monocrotophos, carbofuran, triazophos, imidacloprid, likely leading to a high lethality to natural enemies[70]. A low abundance of natural enemies on aphids might also be attributed to decreased ecosystem service values as a result of the intense agricultural use and simplified landscape diversity that may have decreased insect biodiversity[71]. In contrast, the decreasing input of insecticides and less intense land use in Europe potentially allowed natural enemies to sustain populations that suppressed aphids[32]. Therefore, different use of chemical insecticides and landscape intensity between China and Europe could indirectly affect aphid populations via changing natural enemy efficacy.

Increased nitrogen applications to crops potentially positively increased herbivore populations by improving plant nutrients[72,73]. We found general contrasting trends of N fertilizer applications per area in wheat or cereals between China and Europe over the last five decades. Unfortunately, we had no data of nitrogen content in wheat leaves and stems over large temporal and spatial scales available for performing statistical analyses which could help to explicitly unveil the effects of N fertilizers on aphid abundance. The consistent increase of N fertilizers in China might have contributed to the increase of wheat aphid populations but further study is needed to provide strong evidence. Therefore, we may need to reconsider the role of climate warming

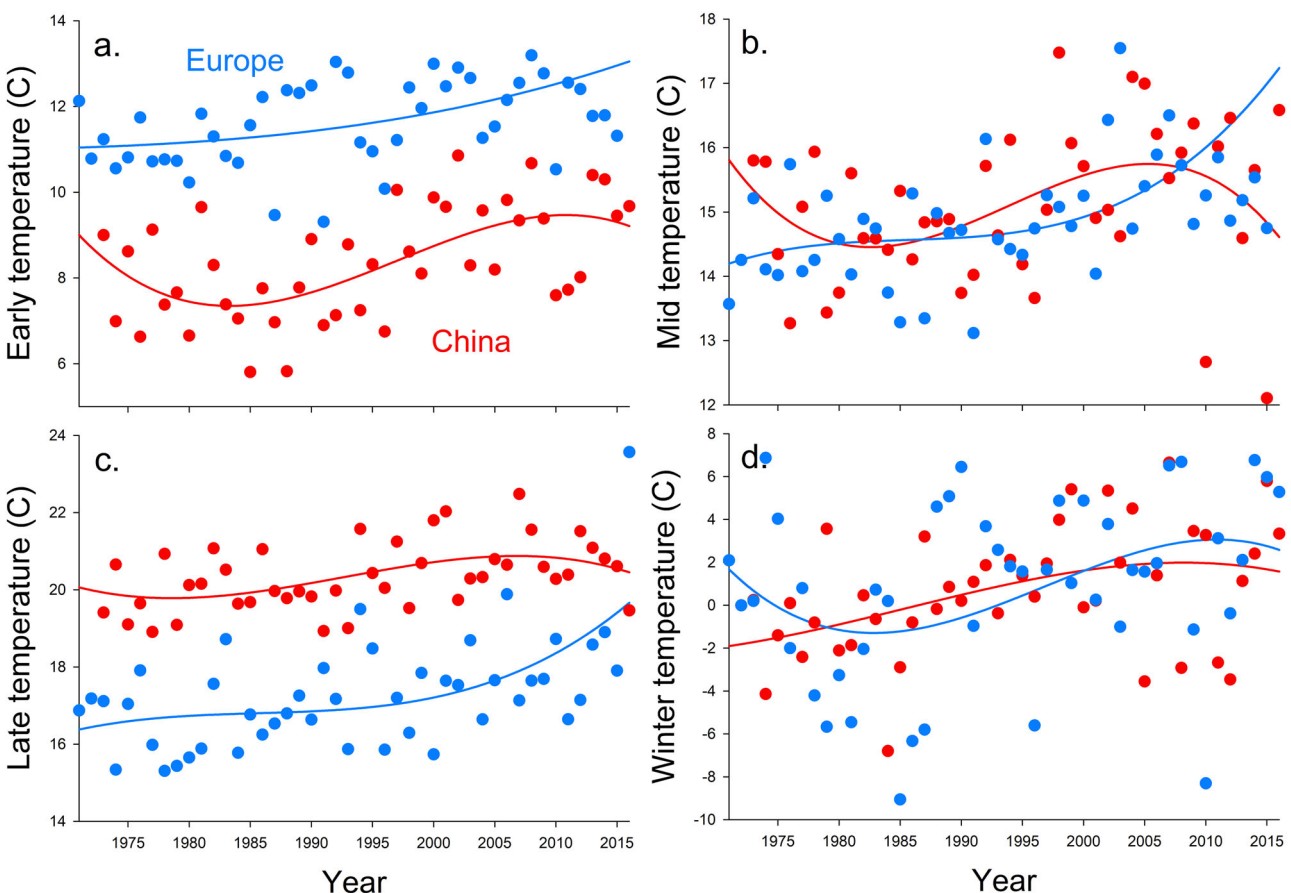

**Fig. 4 Historical temperatures in China and Europe.** Trends of temperatures in China and Europe in early (**a**), mid (**b**), and late (**c**) wheat growing seasons and winter (**d**) from 1971 to 2015. Red lines represent the average for provinces that provided aphid data for China and blue lines represent the average for countries that provided aphid data for Europe.

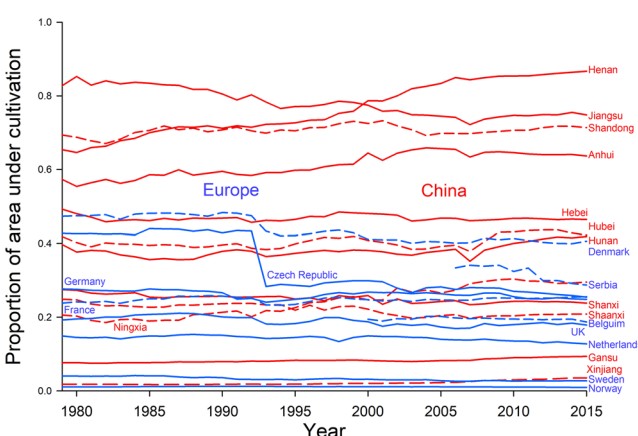

**Fig. 5 Trends of land use in Europe and China.** Provinces in China (red) or countries in Europe (blue) with significant positive patterns of the proportion of land in agricultural use over time were Anhui, Henan, Hubei, Hunan, Ningxia, Shandong, Xinjiang, and France and those with significant negative patterns were Jiangsu, Shaanxi, Shanxi, Czech Republic, Denmark, Germany, Netherlands, Serbia, and Sweden.

in increasing aphid populations and promoting aphid outbreaks in natural environments. Increasing temperature may increase aphid development rates and populations, however, these effects may be counteracted by other environmental factors that are related to temperature. Thus, long-term field data could explicitly

reveal the true net effects in the context of climate warming. Moreover, evaluating the effects of climate change must employ broad geographic scales, because results solely relying on regional data may be biased (in this case, only consideration of the China data would have showed positive effects of warming). In this context, analyses of long-term data with multiple factors in large geographic scales can help elucidate the driving factors on aphid population dynamics.

We note that we could not fully explore all the factors that affect wheat aphid abundance in this study. For example, we did not include other climate variables such as precipitation, because evaluating the net effect of precipitation on insects over multi-decade years may be complicated[74]. In addition, specific management practices, such as tillage, weeding, crop rotations, and wheat variety could be important drivers of aphid abundance, which may also account for variation between China and Europe. Because of limited data, we collected fertilizer and insecticide data for more than wheat cultivation in Europe and of pesticides for more than wheat cultivation in China. Yet, we hypothesize that fertilizer, insecticide, and pesticide applications for all crops are positively related to applications for wheat. To date, various strategies have been employed for aphid-resistant breeding which may impact long-term aphid loads in wheat. But, conventional breeding aphid-resistant wheat cultivars for minimizing the use of insecticides have brought little success[75]. In this study, land use was only assessed at a general level and at a coarse spatial scale. More disaggregated classification and precise criteria such as agricultural-field edge density effects[76] will provide more insights into how to effectively and successfully implement ecological

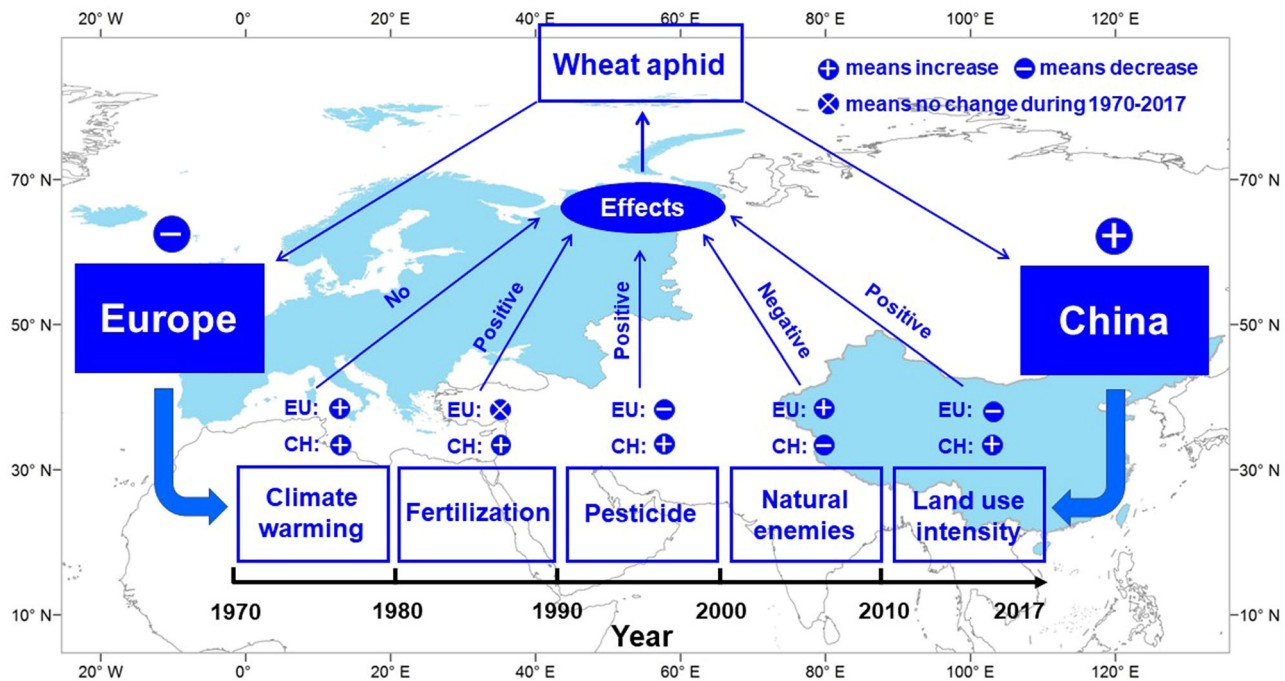

**Fig. 6 Conceptual overview of the variation of wheat aphid and its associated factors between China and Europe during 1970 to 2017.** This map was created by using the ArcGIS 10.3.1 software (ESRI, Redlands, CA, USA, http://www.esri.com/).

intensification strategies based on sustainability in agroecosystems. Moreover, we were unable to include wheat aphid data in North America or Australia because we could not collect enough papers in North America or Australia, so we did not integrate the aphid data by long-term series to analyze the variation trend of wheat aphid by the year.

Our evaluations of the effects of environmental factors on wheat aphid loads in China and Europe have both theoretical and policy implications for developing healthy agroecosystems through pest management under global change. First, consideration of biocontrol ecosystem services by landscape use intensity is critical in wheat pest management[77]. This has become more important under the current drastic land use change. As a significant result of the policy of the "Conversion of degraded farm land into forest and grass land" implemented in China since 1999, China's increasing forest-grassland cover in the last decade[78] may assist to restore the ecosystem functions for biocontrol in the surrounding agricultural fields now and in the future. However, because wheat is so important for food security and the major wheat production regions are outside of China's increasing forest-grassland areas, further policies specifically focused on wheat ecosystems may also be needed to maximize ecosystem values for wheat yield and grain quality. Second, as expected, we showed increasing aphid loads were associated with overuse of chemical insecticides and nitrogen fertilizers in China. The Chinese government has approved the "Double Reduction Plan (reduction of the use of chemical insecticides and fertilizers) at the national level for agro-ecological sustainability since 2016.

Previous studies showed that agricultural practices and climate warming influenced aphid population dynamics. Our multidecadal, continent-level analysis highlights that agricultural practices impact wheat aphid loads more than climate warming. In fact, these long-term data suggest that climate warming may not be an important driver of agricultural pest loads. Our study demonstrates the need to consider policies that reduce the overuse of chemical insecticides and nitrogen fertilizers in agroecosystems. Therefore, under global environment change, consideration of

multiple factors at large spatial-temporal scales will provide more insights for developing effective agroecosystem management to safeguard world food security.

## Methods
**Population dynamics of wheat aphids**. We used three databases (Web of Science, Google Scholar, and China National Knowledge Infrastructure [CNKI]) to search for studies on populations of wheat aphids between January 1970 and December 2017. We used sets of keywords for study collection to identify the relevant articles: (aphid) and (population OR abundances OR dynamics OR long-term OR time series OR observation) and (wheat). We used the following criteria to screen studies in the dataset: (1) the study was a field survey in open wheat plots, (2) the papers reported aphid data for specific dates within a year (i.e., papers reporting a date range within a year or averages of multiple years were excluded), (3) the data of aphid abundances were reported with specific units (per tiller or per m²) so that they could be expressed as aphid densities per tiller, (4) the data include all the aphid species in the study field and not only a single focal species of interest. We excluded data from treatments or studies that reported insecticide application. We converted values given as # aphids $m^{-2}$ to # of aphids 100 tillers$^{-1}$ using three scenarios (400, 650, 900 tillers $m^{-2}$)[5,79]. We performed all analyses with the data from these three scenarios and obtained the same qualitative results and so we only report the middle scenario. Finally, a total of 2141 data points from China and 1169 data points from Europe in 120 articles were collected for wheat aphids (Fig. 7). We expressed population estimates as log (total aphids100 tillers$^{-1}$) to control for right skewed distributions.

**Population dynamics of natural enemies**. We conducted a literature search (published between January 1970 and December 2017) of natural enemies using the keywords to collect the relevant articles from the three databases used for aphid population collections: (aphid) and (natural enemy* or predator* or parasite*) and (population OR abundances OR dynamics OR long-term OR time series OR observation) and (wheat). Finally, a total of 508 data points from China and 123 data points from Europe in 30 articles were collected concerning the population of natural enemies. Data included lacewings, ladybirds, midges, hoverflies, and spiders or the broader categories parasitoids, predators, or total natural enemies. We expressed population estimates as number of enemies 100 tillers$^{-1}$ because the data were normally distributed.

**Temperature**. We obtained historical records on temperature data for 1970–2016 for the provinces in China and countries in Europe that contributed aphid data. The Chinese temperature data were collected from the United States National Oceanic and Atmospheric Administration[80], while European temperature data were collected from the World Bank[81]. The records for China were daily averages

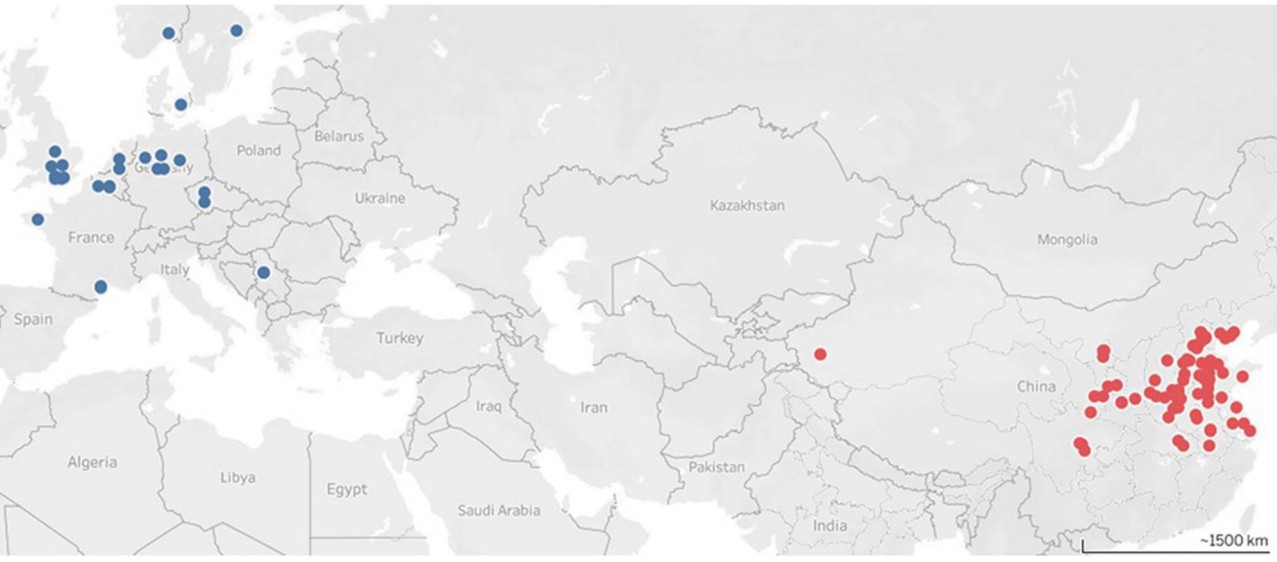

**Fig. 7 Site locations.** Geographical distributions of wheat aphids included in this study of wheat aphid loads in China and Europe. Red circles (2141 data points) represent studies from China and blue circles (1169 data points) represent studies from Europe.

for a single station in each province from which we calculated monthly averages (stations are listed in Supplementary Fig. 4). The records for the European countries were monthly averages aggregated at the country level in the database.

**Fertilizer and pesticide**. We obtained nitrogen fertilizer application rates for wheat (1980–2015) and pesticide (all types) application rates for all crops combined (1991–2015) from the National Bureau of Statistics of China. We obtained data on nitrogen fertilizer application rates for Western Europe crops (biennial 1980–2010) (https://www.fao.org/faostat)[11], for cereal crops in the UK (biennial 1984–2014) from the British Survey of Fertilizer Practice (GOV.UK) and for crops in countries in Europe that contributed aphid data (2002–2015) from FAO (https://www.fao.org/faostat)[11]. We obtained data on insecticide application rates for all crops combined (1991–2015) for countries in Europe that contributed aphid data from FAO (https://www.fao.org/faostat)[11].

**Land use**. We obtained yearly data on areas of land used to grow crops (wheat and cropland) for countries in Europe that contributed aphid data (1979–2015) from FAO (https://www.fao.org/faostat)[11]. We obtained yearly data on areas of land used to grow crops (wheat and cropland) for provinces in China that contributed aphid data (1979–2015) from the China Rural Statistical Yearbooks. We considered the proportion of land under cultivation as a measure of land-use intensity. We calculated the proportion of area under cultivation (area of all combined crops/total area of contributed province or country) and proportion of area under wheat cultivation (area of wheat/total area of contributed province or country). Details of crop categories for Europe and China are in Supplementary Data 3.

**Statistical analysis**. We used regression analyses (H_0: slope = 0) to examine how the abundance of aphids depended on year separately for each ~ten-day period (March: 1–10, 11–20, 21–31; April: 1–10, 11–20, 21–30; May: 1–10, 11–20, 21–31; June: 1–10, 11–20, 21–30; July: 1–10, 11–20, 21–31) in which aphids were present at least at some of the sites (March to May for China and May to July for Europe). To control for multiple points from the same study, we used the average value for each study for each year and ~ten-day period. We performed another set of regressions to examine how the abundance of natural enemies depended on year separately for each ~ten-day period in which they were present at least at some of the sites in China (April to May). In these analyses, we treated each ten-day time period separately and used averages for each year and ten-day period. We used the slopes of regressions of aphid loads and year in models including a random term for source paper to test for differences between the temporal pattern of change over years between China and Europe overall and among early, mid and late season aphid abundances in China and Europe. We considered an effect to be significant when the 95% CI did not overlap zero and we considered two intervals to be different when their 95% CIs did not overlap. We used a funnel plot and Egger's test to examine potential publication bias that may have biased our results.

We tested whether a country or provinces rate of change in aphid loads was correlated with its rate of change in temperature. We performed a mixed model ANOVA to test whether the proportion of area under cultivation depended on year for provinces in China vs. countries in Europe and used slope contrasts to test whether the slopes for China vs. Europe differed on average. We also tested whether slopes for individual countries or provinces differed from zero.

**Reporting summary**. Further information on research design is available in the Nature Research Reporting Summary linked to this article.

## Data availability
Source data behind the graphs can be found in Supplementary Data 1.

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

## Acknowledgements

We would like to thank Min Song, Luqian Li, Jiayin Dai, Yifan Chen, Shanyu Liu, and Yi Zhong for data extraction. This work was supported by National Key Research and Development Program (2017YFD0200600), Program for Science & Technology Innovation Talents in Universities of Henan Province (22HASTIT039), Young Talent Support Project of Henan Province (2021HYTP034), and National Postdoctoral Program for Innovative Talents (BX201700069).

## Author contributions

J.D. conceived the idea. J.D., X.S., and E.S. designed the study. X.S., Y.S., L.M., Z.L., Q.W., D.W., C.Z., and E.S. collected the data. E.S., X.S., Y.S., and L.M. assembled the data. E.S., X.S., and H.Y. analyzed the data. X.S., J.D., and E.S. wrote the manuscript.

## Competing interests

The authors declare no competing interests.
