## [Peer Review File · Communications Biology]

Reviewers' comments:

Reviewer #1 (Remarks to the Author):

The idea of the paper is very interesting and promising, the goal of the paper is original. The authors want to test the effect on temperature, fertilizer and pesticide uses and the area of croplands on aphid loads and their natural enemies in China and Europe on the long run with a meta-analytic approach. They identified some correlations between such covariates. However, I have many concerns about the statistical approach which, to me, is not sufficiently robust to tackle the complexity of the data and properly test the hypothesis. I suspect some confounding effects between covariates that could affect conclusions. I have provided many suggestions in the following this review. In addition, I suggest the authors to provide the data and more details about the methodology employed to obtain them. To finish with a positive point, the paper is well written. For all these reasons, I suggest the authors to make analyses deeper and to resubmit the paper.

Title

Not sure that the title is very well adapted. Maybe that you should mention fertilizer and pesticide (or insecticide) in it.

Introduction

L103-109. Please clearly state that you performed meta-analyses.

Clarify if you are interested by insecticides only or pesticides.

The introduction is a bit long but I think you should include a section the effect of landscape composition (notably the effect of the proportion of cultivated area) on aphid loads with their potential consequences on concentration-dilution effects.

Maybe that you should prioritize the expected effects of the different explanatory covariates on aphid loads to reshape your introduction.

Methods

L131 : « specific dates within a year”, what does it mean ?

L132 : Did you only select aphid abundances as response variable ? If not, please modify the term “abundance”

L133. Please the 4th point is not clear to me.

L133-134. In this moment, it is very not clear why you excluded these treatments. Please justify. I understood the importance of this choice too far after in the ms.

L139. Figure 1. Please provide the number of points from both regions in the caption and add a scale in the figure.

L150. You never mentioned “seasonal patterns” before this sentence. It is confusing about what I expect from the study. Specifically, you did not mention that point in the L125-139 section. If it is not necessary, please remove this or justify before this sentence.

L154-155 : Why did you choose several sources for temperature data ?

L156. Please precise from which places came European temperature data and precise how they have been aggregated.

L152-157 : Please provide the data.

L159-166. Why did you retrieve data from “all crops combined” while you focused on aphid wheat ? I have the same concern for L168.

L168-172. It is very not clear how you calculated land-use intensity. In detail, what kind of data did you extract from FAOSTAT and at which scale such data were considered? Did you aggregate the data? Did you collect data at the annual scale? What kind of landscape element did you include as “land under cultivation” (meadow)? Maybe that you can include this in the suppl. Mat. but details are sadly lacking in the section. It is important because we can suspect that a lack of precision could affect the influence of land use on aphid loads.

L174. Why did you log transformed the data? Did you already try to treat abundances with Poisson models?

L175. Please precise, as you did it at the beginning section to which dates correspond the 10-days periods.

L181. Please mentioned the number of slopes that you used for this test. Please clearly write what is the additional value of this test in comparison to the previous one. After reading the results, I finally understand that you tried to highlight general trends.

I have many concerns about the statistical analysis section :

Please write, as least in the supplementary material the models.

Please clarify that you developed models for Europe and China independently.

Write the number of observations used to run each model.

You did not include random terms while I see potential dependence in your data : from authorship, from locations and years (you need to test spatial and temporal autocorrelation). It will allow to decide which kind of models have to be employed.

You never mentioned fertilizer and insecticide data in this section, how did you include the information in the models ?

I also suggest you to write the hypothesis and the associated model(s).

I wonder if you would not correct your 10-day period by the number of degree-day for each country. I am less familiar with China temperatures but in Europe measuring aphid loads in March in the South of France does not have the same meaning that in Sweden. A minima, you should test the effect of this clustering on the results.

Did you check model residuals?

Moreover, as you employed a meta-analytic approach, some packages could be very useful to you such as the package metaphor and several papers introduce a checklist necessary to validate meta-analyses (Koricheva et al., 2014 or Gurevitch et al., 2018 for example).

Results

L193-200 : Maybe that the difference of aphids between Europe and China came from differences of latitude ranges between areas (as I mentioned in the method section).

I think that the proportion of wheat in the landscape should be included as a covariate in the models

L201 : Not sure that the title of this section is well chosen because most of the paragraph is about aphids. If you have not enough data to test natural enemy loads in Europe, it would be appreciable to mention this point in this section.

L203-207: Please precise which data are used to detect these points. Which periods? Number of observations ?

L207 : Precise why you presented a polynomial curve while this effect is not tested in the models ?

L207-209 : How did you aggregate the data presented in the figure S2 ?

L211 : Please clarify why sometimes you present data on insecticides and sometimes on pesticides (all types).

L211-215 : Instead of doing parallel correlations according to time for inputs and arthropods, why did not you directly test the effect of both pesticide (or insecticide and fertilizer) on arthropod loads ?

L210-221 : To me, if you want to examine the effect of pesticide or fertilizer uses on aphid loads, you need to correct their level of use by the land cover under wheat or at least cereal at the plot or regional scale.

L224 : Why did you choose to represent the winter period on these figures ? If you have hypothesis about this, notably carry-over effects, please clarify it.

L229:239 : I think this section is extremely important to understand relationships between all the compartments of your complex system. As I previously suggested, the area under cultivation (notably the area under wheat cover) need to be related to the levels of pesticide and fertilizer uses. In addition, all such metrics need to be included as explanatory variables in models to explain aphid and natural enemy loads.

I suggest the authors to reshape statistical analysis section and check correlations between all their explanatory covariates (temperature, cropland, inputs) et to evaluate their relative effect on aphids and natural enemy loads. I suggest to prioritize the relationships between variables to eventually (not sure that is the best but maybe to try) perform a structural equation modelling. Indeed, natural enemy can be response as well as explanatory covariates of aphid loads. The paper would also benefit to complexify their models to consider potential data dependence.

Why only temperature? What is the correlation with other climate variables like precipitations

Discussion.

L241-250. As I already said just above, statistical analyses need to be reshaped before concluding.

L263 : Could you please discuss your results about natural enemies in regard to recent papers showing a decrease of arthropod biodiversity within agricultural landscapes in Europe ? Why did you observe a different trend (Sanchez-Bayo et al., 2019; Hallman et al., 2017; Seibold et al., 2019) ?

L265-266 : Could you check this assumption?

L277. Please discuss potential mechanisms linking aphid loads and N fertilizer via yield but also via vigour increase.

L286-304. As your title focused on the relationship between temperature and aphid loads, I suggest this section should be positioned at the beginning of the discussion.

L315 : Yes, you can not include all the factors that possibly impact aphid abundance. However, maybe that you evaluate the correlation between the variables (precipitations...) mentioned in this section and your explanatory covariates.

L325 : If we had data at the regional scale in the study, maybe that you could include data from North America and Australia and consider the latitude as covariate as well as the other explanatory covariates

Reviewer #2 (Remarks to the Author):

The authors compare aphid abundance in ten-day periods over 49 years in wheat fields in both Europe and China using data from previously published papers, natural enemy abundance is included when available. They also assess global warming- based on mean temperatures, fertilizer use and pesticide use over the same period. Aphid abundance increases over the sampling period in China, but not in Europe, temperature changes do not seem to correlate with this difference, but differences in fertilizer and pesticide use also increase over the sampling period in China. This is quite an impressive analysis, but the authors should be cautious about some of the conclusions and additional analyses are suggested. For all univariate linear regressions the full equation should be given along with the N. More justification is needed for splitting the aphids into ten-day periods vs just using either peak abundance for each replicate or mean for each field over the season. Why did you use mean monthly temperature here? Is that really what we expect to influence aphid abundance? What does the literature say specifically about mean temperatures and aphid abundance? Consider aphid abundance vs timing of peak with relationship to temperature. More justification is needed for this. A mixed model incorporating all factors may lend more credence to the conclusions of Figure 7. Chinese provinces or provinces in China- throughout MS. More citations are needed to support the discussion section.

L 15 change to "influences" or "has a large influence on" (as temperature is not the only factor determining insect abundance)

L19 "affecting" rather than determining, for the same reason

L66 specify "herbivorous insect" development

L60-62 reference to some work that has been done in this area e.g. (Gagic et al. 2021, Paredes et al. 2021)

L89 cite

L93 temperatures have

L133 aphids

L133-134 why were studies that reported insecticide application excluded?

L175 delete tilde (~)

L174-176 did you do regressions on both ten-day average aphid abundance and yearly average? This is a bit unclear. For ten-day average models in which multiple samples were taken per field-year was field included as a random effect (repeated measures)?

L176 more justification needed for the different time periods used in China vs Europe- perhaps references to other literature that indicates the time period for peak aphid abundance in wheat in each continent

L 207-209 the assertion that ladybirds and hoverflies (Syrphids in the figure- stay consistent), specifically decline in China is driven for both groups by a single data point in the early years placing doubt on this claim. For Europe the claim about coccinellid abundance over time is not well supported since you only have one data point after the early 90's- I suggest removal. More caution is needed in conclusions from these data. Additionally, if you want to separate coccinellids and syrphids in the text a regression equation should be used to support it.

L218 dose of an individual application or total inputs generally for insecticide? A bit unclear

L220-221 aphid loads.... This is repetition of results from above. The comparison should be in the discussion

L225 font mismatch

L253 "syrphids", since everything else is plural

L255 "have shown" rather than "confirmed"

L258 certainly there published papers that either support this statement or not including one from your source data (Hopper et al. 1995)- also e.g. (Östman et al. 2003, Snyder and Ives 2003, Latham and Mills 2010)

L266 do any citations support this?

L269-272 careful you measure abundance but reference biodiversity, these are not necessarily related. Furthermore some natural enemies- coccinellid for example, have been shown to respond well to intensified landscapes- e.g. (Emery et al. 2021)

L272-274 citations to support this

L275 careful, you didn't measure landscape diversity

L276 changing the natural enemy community? Abundance? Efficacy?

L287-290 since you reference several studies add the appropriate citation- especially for the study conducted over several seasons vs the others

L303-304 not sure the point of this sentence

L305 revisit the role? Unclear

L343 mismatching font

L347 careful here, again, you only looked at mean temperatures (warming), not other aspects of climate change

Fig 2 many of these look nonlinear and may be similarly humped in both China and in Europe, for the periods in which there is enough data

Fig 7 is a bit confusing since you show the trend by continent and then the relationship with aphid abundance, but you never actually regress these factors against aphid abundance- either individually accounting for year and continent as fixed effects and study as a random effect or in a multivariate mixed model with all factors included. Additionally, you don't measure effects here, but correlations, so caution should be taken in the development of this figure- perhaps abundance is more appropriate.

Table S1 Data source table- are the N's the number of field-year replicates or the number of times of one replicate was checked per 10 day period? Did you control for study source in the analysis?

References

Emery, S. E., M. Jonsson, H. Silva, A. Ribeiro, and N. J. Mills. 2021. High agricultural intensity at the landscape scale benefits pests, but low intensity practices at the local scale can mitigate these effects. *Agriculture, Ecosystems & Environment* 306:107199.

Gagic, V., M. Holding, W. N. Venables, A. D. Hulthen, and N. A. Schellhorn. 2021. Better outcomes for pest pressure insecticide use, and yield in less intensive agricultural landscapes. *PNAS* 118:1–6.

Hopper, K. R., S. Aidara, S. Agret, J. Cabal, D. Coutinot, R. Dabire, C. Lesieux, G. Kirk, S. Reichert, F. Tronchetti, and J. Vidal. 1995. Natural enemy impact on the abundance of *Diuraphis noxia* (Homoptera: Aphididae) in wheat in Southern France. *Environmental Entomology* 24:402–408.

Latham, D. R., and N. J. Mills. 2010. Quantifying aphid predation: The mealy plum aphid *Hyalopterus pruni* in California as a case study. *Journal of Applied Ecology* 47:200–208.

Östman, Ö., B. Ekbom, and J. Bengtsson. 2003. Yield increase attributable to aphid predation by ground-living polyphagous natural enemies in spring barley in Sweden. *Ecological Economics* 45:149–158.

Paredes, D., J. A. Rosenheim, R. Chaplin-Kramer, S. Winter, and D. S. Karp. 2021. Landscape simplification increases vineyard pest outbreaks and insecticide use. *Ecology Letters* 24:73–83.

Snyder, W. E., and A. R. Ives. 2003. Interactions between specialist and generalist natural enemies: parasitoids, predators, and pea aphid control. *Ecology* 84:91–107.

Reviewer #3 (Remarks to the Author):

An extensive meta-analysis of the roles of climate and agricultural practices on wheat aphid populations across various regions of China and countries of Europe. Report while temperature has of course increased in both regions, agricultural practices were identified as the significant contributor to differences in aphid populations. It is important to recognise the effect of agricultural practices on aphid populations and identify those which contribute to increase and thus be better able to manage these pests.

Unusually, the potential role of chemical fertilizers is touched on and while there is limited information presented it raises the relevance of fertilizers to be included in future studies.

Pesticide use and land use impacts are included in analysis. L.77 That use of synthetic insecticides can lead to pest outbreaks due to decrease in natural enemy abundance is well known Please also comment on the potential role of pesticide resistance in pest, including aphid, outbreaks. Insecticide resistance can impact aphid populations eg see Gong et al (2021) Chemosphere 269:128747 and Wei et al (2019) Genes 10: 951

The manuscript addresses an important issue -that of teasing out potential factors underlying pest population abundances with a view to limiting their impact on food security. It is a comprehensive attempt to identify potential drivers. There are some issues with the authors being too general or non specific in combining information in different studies-see some specific comments below
Some information as to species of 'wheat aphids' are under discussion. I am not an expert but at least 3 aphid taxa (*Sitobion avenae*, *Diuraphis noxia*, *R. padi*) come up when I look for wheat aphids. Also please comment on the role of aphid resistant strains of wheat. Are these common and potentially useful?

L149 "natural enemies data was too sparse to examine seasonal patterns" please add a sentence to explain how European data was used see results LI 206-7

P italics and spaces around equality and inequality are inconsistent

P is commonly reported to 3 dp and the minimum value limited to 0.001

Re 'land use'

L168 please expand explanation of landuse 'considered proportion of land under cultivation as a measure of land use intensity' Please provide more information as to how this is assessed. Was it not possible to use proportion of non crop or some similar measure which might have more direct relevance to planning?

L 248 assessed natural enemy abundance this is not really a synonym for 'biocontrol services' (L201 Ff) please use ne abundance consistently

L305 it is too general to make conclusions regarding 'pest populations. Need to insert aphid here (and elsewhere eg L 347) that these comments refer to aphid pest populations.

L330 'consideration of biocontrol ecosystem service by landscape diversity is critical in wheat pest management' needs elaboration and referencing as this study does not identify a role for landscape diversity but rather 'proportion of land under cultivation' which is not the same thing.

Responses to reviewers' comments

We have carefully considered reviewers' comments. In the following we address reviewers' comments, point by point. Our responses are in **RED**.

Reviewers' comments:

Reviewer #1 (Remarks to the Author):

The idea of the paper is very interesting and promising, the goal of the paper is original. The authors want to test the effect on temperature, fertilizer and pesticide uses and the area of croplands on aphid loads and their natural enemies in China and Europe on the long run with a meta-analytic approach. They identified some correlations between such covariates.

However, I have many concerns about the statistical approach which, to me, is not sufficiently robust to tackle the complexity of the data and properly test the hypothesis. I suspect some confounding effects between covariates that could affect conclusions. I have provided many suggestions in the following this review. In addition, I suggest the authors to provide the data and more details about the methodology employed to obtain them. To finish with a positive point, the paper is well written. For all these reasons, I suggest the authors to make analyses deeper and to resubmit the paper.

Title

Not sure that the title is very well adapted. Maybe that you should mention fertilizer and pesticide (or insecticide) in it.

Answer: The agricultural practices include fertilizer and pesticide (or insecticide). Thus, we changed the title to “Multidecadal, continent-level analysis indicates agricultural practices impact wheat aphid loads more than climate change”.

Introduction

L103-109. Please clearly state that you performed meta-analyses.

Answer: Yes. We have clearly stated that we determined the main drivers of aphid dynamics

by meta-analyses across the two continents over a 47-year period from 1970 to 2017.

Clarify if you are interested by insecticides only or pesticides.

Answer: It would have been ideal to use only data of insecticides specifically applied to wheat. Indeed, the FAOSTAT database provides insecticide data for European countries but the National Bureau of Statistics of China only has data available for all pesticides together. Both report applications to all crops together. We hypothesize that insecticide applications on wheat are positively correlated with both insecticide and overall pesticide applications for all crops. We have provided evidence for this hypothesis in the discussion.

The introduction is a bit long but I think you should include a section the effect of landscape composition (notably the effect of the proportion of cultivated area) on aphid loads with their potential consequences on concentration-dilution effects.

Maybe that you should prioritize the expected effects of the different explanatory covariates on aphid loads to reshape your introduction.

Answer: We have simplified the introduction and combined some paragraphs. We also added a paragraph about the effect of landscape composition on biodiversity and biological control in agroecosystems. Furthermore, we have reshaped the introduction related to climate warming. First, we consider the role of climate warming on aphid populations. Then, we consider the role of insecticide applications because they could both affect the aphid resistance directly and affect biodiversity and biological control of aphids indirectly. N fertilizers and land use are discussed in the following paragraphs.

Methods

L131: “specific dates within a year”, what does it mean?

Answer: When we searched for wheat aphid papers, we found that some data were associated with a date range rather than an exact date (for instance for an entire month). Also, some papers reported an average value for several years. Such data were not useful for our analyses and so we set a limiting condition of “specific dates within a year” which meant that aphid loads were reported for a specific date in a single year. To clarify, we added the additional

explanation “(i.e., papers reporting a date range within a year or averages of multiple years were excluded)”.

L132: Did you only select aphid abundances as response variable? If not, please modify the term “abundance”

Answer: We used any paper that expressed aphid abundances per tiller or per m². We realize now from your comment that “abundance” does not adequately capture that we examined aphid loads as numbers per tiller. We expanded the description to be “(3) the data of aphid abundances were reported with specific units (per tiller or per m²) so that they could be expressed as aphid densities per tiller,”

L133. Please the 4th point is not clear to me.

Answer: The point we were trying to make was that the paper needed to include all the aphid species occurring in that study and not only a single species of interest. We have modified the description to make the 4th point clearer.

L133-134. In this moment, it is very not clear why you excluded these treatments. Please justify. I understood the importance of this choice too far after in the ms.

Answer: One of the important targets for this research is to compare the aphid dynamics in natural environments between Europe and China over the last five decades. Insecticide applications affect the aphid abundance significantly and quickly, so we excluded data from treatments that reported insecticide applications.

L139. Figure 1. Please provide the number of points from both regions in the caption and add a scale in the figure.

Answer: We have provided the number of data points from both regions in the figure legend: “Red circles (2141 data points) represent studies from China and blue circles (1169 data points) represent studies from Europe”. Adding a scale is somewhat complicated. With such a large span of latitude, there is no single east/west scale that applies to the more northern and more southern parts of the figure simultaneously. So, we have added a scale as requested but

also would like to note that we would be glad to add an explanation regarding the application of the scale (i.e., that it applies to that latitude and cannot apply to the entire figure) if you think this is not something a reader would immediately recognize.

L150. You never mentioned “seasonal patterns” before this sentence. It is confusing about what I expect from the study. Specifically, you did not mention that point in the L125-139 section. If it is not necessary, please remove this or justify before this sentence.

Answer: Yes. We have removed the sentence about seasonal patterns.

L154-155: Why did you choose several sources for temperature data?

Answer: When we collected the temperature data of China, we found that the database from National Oceanic and Atmospheric Administration (NOAA) was systematic and more comprehensive than other databases. For Europe, the database from the World Bank was more systematic than the NOAA. Thus, the temperature data of China were from NOAA and data of Europe were from World Bank.

L156. Please precise from which places came European temperature data and precise how they have been aggregated.

Answer: The European temperature data were already aggregated by country and month in the World Bank dataset. So, we did not need to aggregate them ourselves. We have revised the language of the text to make this clear.

L152-157: Please provide the data.

Answer: We have added a table S2 that has these data.

L159-166. Why did you retrieve data from “all crops combined” while you focused on aphid wheat? I have the same concern for L168.

Answer: Data of fertilizer, pesticide and insecticide applications were not always available specifically for wheat. Rather, we found that the data sources were sometimes limited in the databases. The database of the National Bureau of Statistics of China provides fertilizer data

for wheat per area but only provides pesticide data for all crops. No data were available for insecticides alone for wheat or for wheat together with other crops. Similarly, the FAOSTAT database only provides fertilizer and insecticide data for Europe crops as a group, rather than for wheat alone. We hypothesize that insecticide applications on wheat are positively correlated with both insecticide and overall pesticide applications for all crops. We have provided evidence for this hypothesis in the discussion.

L168-172. It is very not clear how you calculated land-use intensity. In detail, what kind of data did you extract from FAOSTAT and at which scale such data were considered? Did you aggregate the data? Did you collect data at the annual scale? What kind of landscape element did you include as “land under cultivation” (meadow)? Maybe that you can include this in the suppl. Mat. but details are sadly lacking in the section. It is important because we can suspect that a lack of precision could affect the influence of land use on aphid loads.

Answer: In FAOSTAT, they have data on area harvested of wheat aggregated by country reported by year. We did not have to aggregate these data. Likewise, there is the option of selecting “crops primary” which reports the total area of crops harvested by country by year. We did not further refine or limit these data. But, land use such as “meadow” is not included. In the China Rural Statistical Yearbooks they also have data for area of wheat and of all crops combined aggregated at the province level by year. We did not need to aggregate those data either and did not change what this source considered to be a crop. Again, land use such as “meadow” is not included.

We have added a new table S3 that lists the crop data details for Europe and China.

L174. Why did you log transformed the data? Did you already try to treat abundances with Poisson models?

Answer: The data scaled orders of magnitude (i.e., were highly right skewed) which made a log transformation the most appropriate transformation. We now specify why we used a log transformation.

L175. Please precise, as you did it at the beginning section to which dates correspond the

10-days periods.

Answer: Added these details.

L181. Please mentioned the number of slopes that you used for this test. Please clearly write what is the additional value of this test in comparison to the previous one. After reading the results, I finally understand that you tried to highlight general trends.

Answer: Added

I have many concerns about the statistical analysis section:

Please write, as least in the supplementary material the models.

Please clarify that you developed models for Europe and China independently.

Answer: We have clarified.

Write the number of observations used to run each model.

Answer: We have added these to Figure 2 and S1.

You did not include random terms while I see potential dependence in your data: from authorship, from locations and years (you need to test spatial and temporal autocorrelation). It will allow to decide which kind of models have to be employed.

Answer: No. We did not include random terms in our models in the original version of our manuscript. We have now repeated the analyses in Fig. 2 with a random term for paper (which is a good summary variable that captures locations and authors) included in the models. Of the five significant relationships, four remain significant (March 21-31, April 1-10, April 11-20 & May 1-10 in China) and one previously significant relationship (March 11-20 in China) becomes non-significant. No previously non-significant relationships become significant. Because the results were so qualitatively similar and the R^2 values from the models with the fixed effect of year are useful, we have kept the analyses the same for Fig 2. We now note the effect of including a random term for paper in the models on the results in Fig. 2.

For Fig. 3, including a term for paper had a more important effect. Mainly, the temporal

change in the early period in Europe becomes significantly different from zero. We did not report R^2 values in that figure and so we have changed the analyses and Fig. 3 to include a random term for paper.

You never mentioned fertilizer and insecticide data in this section, how did you include the information in the models?

Answer: We did not. Those data are not resolved more narrowly than year.

I also suggest you to write the hypothesis and the associated model(s).

Answer: We have added this.

I wonder if you would not correct your 10-day period by the number of degree-day for each country. I am less familiar with China temperatures but in Europe measuring aphid loads in March in the South of France does not have the same meaning that in Sweden. A minima, you should test the effect of this clustering on the results.

Answer: We have repeated the analyses in Fig. 2 with a random term for province (China) or country (Europe) included in the models. Of the five significant relationships, four remain significant (March 21-31, April 1-10, April 11-20 & May 1-10 in China) and one previously significant relationship (March 11-20 in China) becomes non-significant. No previously non-significant relationships become significant. Because the results were so qualitatively similar and the R^2 values from the models with the fixed effect of year are useful, we have kept the analyses the same for Fig 2. We now note the effect of including a random term for country or province in the models on the results in Fig. 2.

Did you check model residuals?

Answer: Yes. They did not indicate problems with our fitted models.

Moreover, as you employed a meta-analytic approach, some packages could be very useful to you such as the package metaphor and several papers introduce a checklist necessary to validate meta-analyses (Koricheva et al., 2014 or Gurevitch et al., 2018 for example).

Answer: Great suggestion. We have added a funnel plot and performed an Egger's test to demonstrate that there was no publication bias. This is now Fig S6.

Results

L193-200: Maybe that the difference of aphids between Europe and China came from differences of latitude ranges between areas (as I mentioned in the method section).

Answer: See above for response.

I think that the proportion of wheat in the landscape should be included as a covariate in the models.

Answer: We already examine the effect of land use change in subsequent analyses. The purpose of this set of analyses was to examine the temporal patterns. These later analyses examine potential causes such as proportion of wheat in the landscape.

L201: Not sure that the title of this section is well chosen because most of the paragraph is about aphids. If you have not enough data to test natural enemy loads in Europe, it would be appreciable to mention this point in this section.

Answer: We have revised the title of this section to "Abundance trend of natural enemies coupled with wheat aphids". We also have mentioned that we could not collect as much data of natural enemy loads in Europe as in China.

L203-207: Please precise which data are used to detect these points. Which periods? Number of observations?

Answer: We have clarified the information of the observations, periods and number of data points for natural enemies in China and Europe.

L207: Precise why you presented a polynomial curve while this effect is not tested in the models?

Answer: These polynomial curves are only for visualization. We have added this information to the figure legend.

L207-209: How did you aggregate the data presented in the figure S2?

Answer: These are aggregated across all studies within a continent.

L211: Please clarify why sometimes you present data on insecticides and sometimes on pesticides (all types).

Answer: It would have been ideal to use only data of insecticides specifically applied to wheat. Indeed, the FAOSTAT database provides insecticide data for European countries (all crops) but the National Bureau of Statistics of China only has data available for all pesticides together (all uses). We hypothesize that insecticide applications on wheat are positively correlated with both insecticide and overall pesticide applications for all crops. We have provided evidence for this hypothesis in the discussion.

L211-215: Instead of doing parallel correlations according to time for inputs and arthropods, why did not you directly test the effect of both pesticide (or insecticide and fertilizer) on arthropod loads?

Answer: The data intervals (yearly vs. every two years) and data time periods (when do they start and end) differ among variables and there are multiple sources of fertilization data. So, we decided to show the trends of the variables rather than use a direct correlation approach.

L210-221: To me, if you want to examine the effect of pesticide or fertilizer uses on aphid loads, you need to correct their level of use by the land cover under wheat or at least cereal at the plot or regional scale.

Answer: The units for N fertilizer use are already kg/ha/yr. Pesticides are not only on wheat and so we cannot scale those values to per area of wheat or cereal cultivation.

L224: Why did you choose to represent the winter period on these figures? If you have hypothesis about this, notably carry-over effects, please clarify it.

Answer: Yes, winter temperature can influence insect demography through a series of scenarios. Warmer temperatures in winter could increase aphid mortality and reduce

reproductive potential by reducing nutritional reserves during the dormant diapause period (Xiao et al., 2017). Across a 20 year period, warmer temperatures in winter affected the emergence of both aphid and its Hymenopteran parasitoid, by advancing attack phenology of parasitoid to aphid (Senior et al., 2021).

We have discussed the carry-over effects of warmer temperature in winter in discussion.

L229-239: I think this section is extremely important to understand relationships between all the compartments of your complex system. As I previously suggested, the area under cultivation (notably the area under wheat cover) need to be related to the levels of pesticide and fertilizer uses.

Answer: The units for N fertilizer use are already kg/ha/yr. We have clarified in every occurrence that the fertilization values are per area.

In addition, all such metrics need to be included as explanatory variables in models to explain aphid and natural enemy loads.

Answer: As we note above, the frequency and date ranges of data vary and so we do not think this is a valuable approach.

I suggest the authors to reshape statistical analysis section and check correlations between all their explanatory covariates (temperature, cropland, inputs) et to evaluate their relative effect on aphids and natural enemy loads. I suggest to prioritize the relationships between variables to eventually (not sure that is the best but maybe to try) perform a structural equation modelling. Indeed, natural enemy can be response as well as explanatory covariates of aphid loads. The paper would also benefit to complexify their models to consider potential data dependence.

Answer: SEM is not practical in this case because the frequency and date ranges of data vary.

Why only temperature? What is the correlation with other climate variables like precipitations?

Answer: The effects of precipitation on wheat aphids are unpredictable that cause a series

direct and indirect impacts to wheat aphids.

Rainfall can impede aphid dispersal and dislodge aphids from host plants directly, potentially leading to mortality from impact, predation, or starvation (Thackray et al. 2004; Winder 1990). Furthermore, extreme precipitation events are predicted to impact terrestrial ecosystems through alterations in resources, such as changes in plant growth and chemical composition, and by disrupting interactions between plants and insects (Knapp et al. 2008).

Precipitation could also affect aphid loads indirectly by trophic networks. Moderate precipitation increases aphid loads by increasing water availability and plant productivity. Excess water can alter plant nutrition and amino acid concentrations (Garten et al., 2009; Johnson et al., 2011), and are considered to have far more disruptive effects on plant-insect interactions (Pritchard et al., 2007). Drought had a negative impact on plant vigor and increased plant concentrations of defensive chemicals, influencing the population dynamics of aphids (Leybourne et al., 2021), with consequences that cascade through trophic networks (Johnson et al., 2011; Rodríguez- Castañeda, 2013).

Thus, we could not include the climate factor of precipitation.

Johnson, S., Staley, J., McLeod, F., & Hartley, S. 2011. Plant- mediated effects of soil invertebrates and summer drought on above- ground multitrophic interactions. *Journal of Ecology*, 99, 57-65.

Rodríguez- Castañeda, G. 2013. The world and its shades of green: A meta- analysis on trophic cascades across temperature and precipitation gradients. *Global Ecology and Biogeography*, 22, 118-130.

Leybourne, D. J., Preedy, K. F., Valentine, T. A., Bos, J. I., & Karley, A. J. 2021. Drought has negative consequences on aphid fitness and plant vigor: Insights from a meta- analysis. *Ecology and Evolution*, 11, 11915-11929.

Thackray, D. J., A. J. Diggle, F. A. Berlandier, and R. A. C. Jones. 2004. Forecasting aphid outbreaks and epidemics of Cucumber mosaic virus in lupin crops in a Mediterranean type environment. *Virus Research* 100, 67-82.

Winder, L. 1990. Predation of the cereal aphid *Sitobion avenae* by polyphagous predators on the ground. *Ecological Entomology*, 15, 105-110.

Knapp, A.K., Beier, C., Briske, D.D. et al. 2008 Consequences of more extreme precipitation regimes for terrestrial ecosystems. *BioScience*, 58, 811-821.

Garten, C. Jr., Classen, A., and Norby, R. 2009. Soil moisture surpasses elevated CO₂ and temperature as a control on soil carbon dynamics in a multi-factor climate change experiment. *Plant & Soil*, 319, 85-94.

Johnson, S. N., Staley, J. T., McLeod, F. A. L., and Hartley, S. E. 2011. Plant-mediated effects of soil invertebrates and summer drought on aboveground multitrophic interactions. *Journal of Ecology*, 99, 57-65.

Pritchard, J., Griffiths, B., and Hunt, E. J. 2007. Can the plant-mediated impacts on aphids of elevated CO₂ and drought be predicted? *Global Change Biology*, 13, 1616-1629.

Discussion.

L241-250. As I already said just above, statistical analyses need to be reshaped before concluding.

Answer: Yes. We have now incorporated random effects to account for study as well as province (China) or country (Europe) as suggested.

L263: Could you please discuss your results about natural enemies in regard to recent papers showing a decrease of arthropod biodiversity within agricultural landscapes in Europe? Why did you observe a different trend (Sanchez-Bayo et al., 2019; Hallman et al., 2017; Seibold et al., 2019)?

Answer: In the discussion, we address the different trends compared to other reports which mainly focused on natural ecosystems, especially grasslands and forests. Farmlands experienced more intense disturbance than the other land types.

L265-266: Could you check this assumption?

Answer: We have verified this assumption that insecticide applications are positively correlated to pesticide applications by some references (Peshin et al., 2009; Zhang et al., 2011).

Peshin, R., & Dhawan, A. K. (Eds.). (2009). *Integrated Pest Management: Volume 1: Innovation-Development Process (Vol. 1)*. Springer Science & Business Media.

Zhang W, Jiang F, Ou J. 2011. Global pesticide consumption and pollution: with China as a focus. *Proceedings of the International Academy of Ecology and Environmental Sciences* 1 125-144.

L277. Please discuss potential mechanisms linking aphid loads and N fertilizer via yield but also via vigour increase.

Answer: Yes. We have reshaped the discussion linking aphid loads and N fertilizer via vigor increases.

“Increased nitrogen applications to crops potentially positively increased herbivore populations by improving plant nutrients (Aqueel and Leather, 2011; Gao et al. 2018)”

Aqueel MA, Leather SR. 2011. Effect of nitrogen fertilizer on the growth and survival of *Rhopalosiphum padi* (L.) and *Sitobion avenae* (F.) (Homoptera: Aphididae) on different wheat cultivars. *Crop Protection* 30: 216-221.

Gao J, Guo HJ, Sun YC, Ge F. 2018. Juvenile hormone mediates the positive effects of nitrogen fertilization on weight and reproduction in pea aphid. *Pest Management Science* 74: 2511-2519.

L286-304. As your title focused on the relationship between temperature and aphid loads, I suggest this section should be positioned at the beginning of the discussion.

Answer: Yes. We have positioned this section at the beginning of the discussion.

L315: Yes, you can not include all the factors that possibly impact aphid abundance. However, maybe that you evaluate de correlation between the variables (precipitations...) mentioned in this section and your explanatory covariates.

Answer: Please see above for the explanation.

L325: If we had data at the regional scale in the study, maybe that you could include data

from North America and Australia and consider the latitude as covariate as well as the other explanatory covariates

Answer: We now note: “Moreover, we were unable to include wheat aphid data in North America or Australia because we could not collect enough papers in North America or Australia, so we did not integrate the aphid data by long-term series to analyze the variation trend of wheat aphid by the year”.

Reviewer #2 (Remarks to the Author):

The authors compare aphid abundance in ten-day periods over 49 years in wheat fields in both Europe and China using data from previously published papers, natural enemy abundance is included when available. They also assess global warming- based on mean temperatures, fertilizer use and pesticide use over the same period. Aphid abundance increases over the sampling period in China, but not in Europe, temperature changes do not seem to correlate with this difference, but differences in fertilizer and pesticide use also increase over the sampling period in China. This is quite an impressive analysis, but the authors should be cautious about some of the conclusions and additional analyses are suggested. For all univariate linear regressions the full equation should be given along with the N. More justification is needed for splitting the aphids into ten-day periods vs just using either peak abundance for each replicate or mean for each field over the season. Why did you use mean monthly temperature here? Is that really what we expect to influence aphid abundance? What does the literature say specifically about mean temperatures and aphid abundance? Consider aphid abundance vs timing of peak with relationship to temperature. More justification is needed for this. A mixed model incorporating all factors may lend more credence to the conclusions of Figure 7. Chinese provinces or provinces in China- throughout MS. More citations are needed to support the discussion section.

(1) Question 1: For all univariate linear regressions the full equation should be given along with the N.

Answer: Added to figure.

(2) Question 2: More justification is needed for splitting the aphids into ten-day periods vs just using either peak abundance for each replicate or mean for each field over the season.

Answer: We were interested in early, middle, and late growing season patterns.

(3) Question 3: Why did you use mean monthly temperature here? Is that really what we expect to influence aphid abundance? What does the literature say specifically about mean temperatures and aphid abundance? Consider aphid abundance vs timing of peak with relationship to temperature. More justification is needed for this.

Answer: We have expanded our justification.

(4) Question 4: A mixed model incorporating all factors may lend more credence to the conclusions of Figure 7.

Answer: This is a figure akin to a graphical abstract. It summarizes our conceptual results. We now specify this in the figure legend.

(5) Question 5: Chinese provinces or provinces in China- throughout MS.

Answer: We have unified to provinces in China and countries in Europe throughout the MS.

(6) Question 6: More citations are needed to support the discussion section.

Answer: We have added more citations in discussion.

L 15 change to “influences” or “has a large influence on” (as temperature is not the only factor determining insect abundance)

Answer: Yes. We have revised “determines” to “has a large influence on”.

L19 “affecting” rather than determining, for the same reason

Answer: Yes. We have revised “determining” to “affecting”.

L66 specify “herbivorous insect” development

Answer: Yes. We have specified to “herbivorous insect”.

L60-62 reference to some work that has been done in this area e.g. (Gagic et al. 2021, Paredes et al. 2021)

Answer: Yes. We now reference these.

L89 cite

Answer: We have cited references to support this point.

L93 temperatures have

Answer: Yes. Revised.

L133 aphids

Answer: Yes. Revised.

L133-134 why were studies that reported insecticide application excluded?

Answer: One of the goals of this study is to compare the aphid patterns in natural environments between Europe and China over the last five decades. Insecticide applications affect the aphid abundance significantly, so we excluded data from treatments or studies that reported insecticide applications. We retain the aphid data from control treatments or field surveys in natural environments.

L175 delete tilde (~)

Answer: It means “approximately” so we have retained it but specify the actual periods (see response to reviewer 1).

L174-176 did you do regressions on both ten-day average aphid abundance and yearly average? This is a bit unclear. For ten-day average models in which multiple samples were taken per field-year was field included as a random effect (repeated measures)?

Answer: Yes, we have done the regressions on both ten-day average aphid abundance (Fig 2) and yearly average (Fig. 3 and Fig. 4A).

L176 more justification needed for the different time periods used in China vs Europe- perhaps references to other literature that indicates the time period for peak aphid abundance in wheat in each continent

Answer: The emergence period of wheat aphid is from March to May in China and from May to July in Europe (Table S1), thus we arranged the regression analyses by the time that March to May for China and May to July for Europe.

We have supplied the justification in statistical analysis.

L 207-209 the assertion that ladybirds and hoverflies (Syrphids in the figure- stay consistent), specifically decline in China is driven for both groups by a single data point in the early years placing doubt on this claim. For Europe the claim about coccinellid abundance over time is not well supported since you only have one data point after the early 90's- I suggest removal. More caution is needed in conclusions from these data. Additionally, if you want to separate coccinellids and syrphids in the text a regression equation should be used to support it.

Answer: We have softened our conclusions.

L218 dose of an individual application or total inputs generally for insecticide? A bit unclear

Answer: Here, it means the total insecticide inputs in Europe. Revised.

L220-221 aphid loads.... This is repetition of results from above. The comparison should be in the discussion

Answer: We have removed the repetitive sentence, and moved the comparison in the discussion.

L225 font mismatch

Answer: Revised.

L253 "syrphids", since everything else is plural

Answer: Yes. Revised, and checked throughout the MS.

L255 “have shown” rather than “confirmed”

Answer: Yes. Revised.

L258 certainly there published papers that either support this statement or not including one from your source data (Hopper et al. 1995)- also e.g. (Östman et al. 2003, Snyder and Ives 2003, Latham and Mills 2010)

Answer: Yes. We now reference those papers.

L266 do any citations support this?

Answer: We have verified this assumption by some references.

L269-272 careful you measure abundance but reference biodiversity, these are not necessarily related. Furthermore, some natural enemies- coccinellid for example, have been shown to respond well to intensified landscapes- e.g. (Emery et al. 2021)

Answer: Yes. We have removed the not-related reference and replaced by the correct reference.

L272-274 citations to support this

Answer: Yes. We have added the citations to support this point.

L275 careful, you didn't measure landscape diversity

Answer: Yes. We have replaced “landscape diversity” with “landscape intensity”.

L276 changing the natural enemy community? Abundance? Efficacy?

Answer: Yes. We have revised by “natural enemy efficacy”.

L287-290 since you reference several studies add the appropriate citation- especially for the study conducted over several seasons vs the others

Answer: Yes. We have added the references over long-term series and short-term series.

L303-304 not sure the point of this sentence

Answer: Because of the difference of climate type between China and Europe, the phenology of wheat aphid in China is different from in Europe. The emergence period of wheat aphid is from March to May in China and from May to July in Europe. Thus, we should consider the effects of climate type and phenology on aphid dynamics.

L305 revisit the role? Unclear

Answer: It's generally accepted that increasing temperature increases pest insect abundance. However, our results showed that there is no positive relationship between temperature and aphid abundance in Europe. Temperature is not as important as previous publications report.

L343 mismatching font

Answer: Yes. Revised.

L347 careful here, again, you only looked at mean temperatures (warming), not other aspects of climate change

Answer: Yes. Revised.

Fig 2 many of these look nonlinear and may be similarly humped in both China and in Europe, for the periods in which there is enough data

Answer: We examined 2nd order polynomials and the fit was not significantly better.

Fig 7 is a bit confusing since you show the trend by continent and then the relationship with aphid abundance, but you never actually regress these factors against aphid abundance- either individually accounting for year and continent as fixed effects and study as a random effect or in a multivariate mixed model with all factors included. Additionally, you don't measure effects here, but correlations, so caution should be taken in the development of this figure- perhaps abundance is more appropriate.

Answer: This is a conceptual figure akin to a graphical abstract which we now make explicit.

Table S1 Data source table- are the N's the number of field-year replicates or the number of times of one replicate was checked per 10 day period? Did you control for study source in the analysis?

Answer: The N represents the data points we collected from this paper averaged within the time period. See above for new analyses with a random term for study source.

Reviewer #3 (Remarks to the Author):

An extensive metanalysis of the roles of climate and agricultural practices on wheat aphid populations across various regions of China and countries of Europe. Report while temperature has of course increased in both regions, agricultural practices were identified as the significant contributor to differences in aphid populations. It is important to recognise the effect of agricultural practices on aphid populations and identify those which contribute to increase and thus be better able to manage these pests. Unusually, the potential role of chemical fertilizers is touched on and while there is limited information presented it raises the relevance of fertilizers to be included in future studies.

Pesticide use and land use impacts are included in analysis. L.77 That use of synthetic insecticides can lead to pest outbreaks due to decrease in natural enemy abundance is well known. Please also comment on the potential role of pesticide resistance in pest, including aphid, outbreaks. Insecticide resistance can impact aphid populations eg see Gong et al (2021) Chemosphere 269:128747 and Wei et al (2019) Genes 10: 951.

Answer: We have commented on the potential role of pesticide resistance in pest outbreaks, especially for aphids.

The manuscript addresses an important issue -that of teasing out potential factors underlying pest population abundances with a view to limiting their impact on food security. It is a comprehensive attempt to identify potential drivers. There are some issues with the authors being too general or non specific in combining information in different studies-see some specific comments below

Some information as to species of 'wheat aphids' are under discussion. I am not an expert but

at least 3 aphid taxa (*Sitobion avenae*, *Diuraphis noxia*, *R. padi*) come up when I look for wheat aphids.

Answer: When we collected the wheat aphid data, we did not distinguish aphid species. The aphid species are quite different between China and Europe. Actually, most of the source papers do not distinguish the aphid species. Thus, we extracted the total number of aphids for the database. We have clarified this point again in Materials and Methods.

Also please comment on the role of aphid resistant strains of wheat. Are these common and potentially useful?

Answer: Yes, we should consider aphid-resistant cultivars of wheat. To date, various strategies have been endeavored for aphids-resistant engineering. However, source genes are limited for aphid resistance. As a result, conventional breeding methods to breed aphid-resistant cultivars for minimizing use of insecticides have brought little success (Yu et al., 2014). We now mention aphid-resistant cultivars in the discussion.

Yu X, Wang G, Huang S, Ma Y, Xia L. 2014. Engineering plants for aphid resistance: current status and future perspectives. *Theoretical and Applied Genetics* 127: 2065-2083.

L149 “natural enemies data was too sparse to examine seasonal patterns” please add a sentence to explain how European data was used see results L1 206-7

Answer: We have added the explanation that “we collected natural enemy loads of 108 data points covering 5 countries during 1992-2007 in Europe. We analyzed the annual trends of natural enemies in Europe.”

Furthermore, as one of the reviewers suggested that the “seasonal patterns” were never mentioned before this sentence, they suggested to remove this sentence (which we did).

P italics and spaces around equality and inequality are inconsistent

Answer: Yes. Revised, and checked throughout the MS.

P is commonly reported to 3 dp and the minimum value limited to 0.001

Answer: Yes. Revised, and checked throughout the text.

Re 'land use'

L168 please expand explanation of landuse 'considered proportion of land under cultivation as a measure of land use intensity'. Please provide more information as to how this is assessed. Was it not possible to use proportion of non crop or some similar measure which might have more direct relevance to planning?

Answer: Agricultural intensification since 1950 has resulted in serious loss of biodiversity and ecosystem function within agricultural landscapes (Benton et al., 2003). Negative effects of the proportion of cultivated land were found on biological control by natural enemies (Rausch et al, 2016). A relative increase of cultivated land from 2% to 100% in the 1 km radius reduced the level of natural pest control by about 46%, suggesting that landscape is a major determinant of pest control in agroecosystems (Tschardt et al., 2005; Meehan et al., 2011). In contrast, low-intensity agriculture enhanced biodiversity and promoted biological control (Macfadyen et al., 2009). To satisfy the food demand of vast populations in China, agricultural fields have long been intensified and landscapes simplified (Liu et al. 2018), likely decreasing ecosystem services in wheat.

We obtained the land use data by searching the keywords of wheat and cropland, respectively, in China and Europe. We searched the data at country level in Europe and at province level in China. The obtained data at annual scale for each province or country to analyze the dynamics of proportion of area under (wheat) cultivation. The landscape element means cropland, not include meadow, forest (see new Table S3).

We have provided more information in introduction and materials and methods to clarify it.

Benton TG, Vickery JA, Wilson JD. 2003. Farmland biodiversity: is habitat heterogeneity the key? *Trends in Ecology & Evolution* 18: 182-188.

Liu J, Ning J, Kuang W, Xu X. 2018. Spatio-temporal patterns and characteristics of land-use change in China during 2010-2015. *Journal of Geographical Sciences* 73: 789-802.

- Meehan TD, Werling BP, Landis DA, Gratton C. 2011. Agricultural landscape simplification and insecticide use in the Midwestern United States. Proceedings of the National Academy of Sciences USA. 108: 11500-11505
- Macfadyen S, Gibson R, Polaszek A, Morris RJ, Craze PG, Planque R, Symondson WOC, Memmott J. 2009. Do differences in food web structure between organic and conventional farms affect the ecosystem service of pest control? Ecology Letters 12: 229-238
- Rusch A, Chaplin-Kramer R, Gardiner M M, Hawro V, Holland J, Landis D, ... Bommarco R. 2016. Agricultural landscape simplification reduces natural pest control: A quantitative synthesis. Agriculture, Ecosystems & Environment. 221: 198-204.
- Tscharntke T, Klein AM, Kruess A, Steffan-Dewenter I, Thies C, 2005. Landscape perspectives on agricultural intensification and biodiversity-ecosystem service management. Ecology Letters 8: 857-874.

L 248 assessed natural enemy abundance this is not really a synonym for ‘biocontrol services’ (L201 Ff) please use ne abundance consistently

Answer: Yes. Revised and checked throughout the MS.

L305 it is too general to make conclusions regarding ‘pest populations. Need to insert aphid here (and elsewhere eg L 347) that these comments refer to aphid pest populations.

Answer: Yes. Revised and checked throughout the MS.

L330 ‘consideration of biocontrol ecosystem service by landscape diversity is critical in wheat pest management’ needs elaboration and referencing as this study does not identify a role for landscape diversity but rather ‘proportion of land under cultivation’ which is not the same thing.

Answer: Yes, the expression of “landscape diversity” is not correct. We have replaced the “landscape diversity” with “landscape intensity”. We also added some references to explain this point.

REVIEWERS' COMMENTS:

Reviewer #3 (Remarks to the Author):

The authors have completed a thorough and comprehensive review of their submission. Responses to all points raised have been clearly articulated to facilitate a satisfactory understanding of responses in review.